# IMPLICIT NEURAL COMPRESSION OF POINT CLOUDS VIA LEARNABLE ACTIVATION FUNCTION

## ABSTRACT

Efficiently compressing and transmitting large-scale high-fidelity 3D point clouds is a critical bottleneck for practical applications. We introduce a novel framework that reformulates point cloud compression as model compression. Our framework models high-fidelity point cloud geometry and attribute with compact implicit neural representations (INR) separately and then compresses the model parameters directly via quantization and entropy coding, decoupling representation from compression. To ensure this neural representation is both faithful and efficient, we employ Kolmogorov-Arnold Network (KAN) as the INR backbone. Thanks to its superior approximation properties and parameter efficiency, KAN can easily capture fine-grained details missed by traditional MLP. Extensive evaluations on datasets such as KITTI, ScanNet, and 8iVFB demonstrate that our method significantly outperforms the MPEG standard and prior implicit neural representation approaches. Notably, it achieves competitive rate-distortion performance against state-of-the-art deep learning codecs. Our findings establish implicit neural compression as a powerful and practical pathway for developing the next generation of high-efficiency point cloud codecs.

## 1 INTRODUCTION

Point clouds have emerged as a foundational data modality for 3D perception, powering critical applications in autonomous driving (Li et al., 2020; Cui et al., 2021), augmented and virtual reality (AR/VR) (Lim et al., 2022; Wang et al., 2023), and embodied intelligence (Qi et al., 2024). The proliferation of advanced LiDAR sensing technologies (Raj et al., 2020) has made the acquisition of large-scale, high-resolution point clouds more feasible than ever.

However, raw point clouds' massive scale, spatial sparsity, and lack of explicit topological structure create substantial memory and bandwidth overhead, which severely impedes their practical deployment (Graziosi et al., 2020). Consequently, the development of efficient Point Cloud Compression (PCC) solutions becomes a critical and pressing necessity.

Early efforts to standardize PCC were developed by the Moving Picture Experts Group (MPEG), resulting in the release of two foundational frameworks: geometry-based PCC (G-PCC) (Schwarz et al., 2018) and video-based PCC (V-PCC) (Graziosi et al., 2020). Deep learning has subsequently driven a paradigm shift in the field. Many of these methods (Huang & Liu, 2019; Quach et al., 2020; Que et al., 2021; Zhang et al., 2024) are built upon generic encoder-decoder architectures, which encode a point cloud into a compact latent representation for subsequent reconstruction. While these data-driven codecs often surpass the rate-distortion (RD) performance of traditional standards, their dependence on large-scale pre-training datasets limits generalization to out-of-distribution data.

To address the generalization limitations of data-driven methods, a new paradigm based on implicit neural representations (INR) has emerged (Xue et al., 2024; Ruan et al., 2024a;b). INR-based methods do not directly learn how to reconstruct point clouds. Instead, they train a lightweight implicit neural representation to model the distribution of a single point cloud in 3D space. This strategy inherently avoids the generalization issues of data-driven codecs. However, how to select the optimal INR backbone for compression remains an open research problem, as it must balance the conflicting demands of capturing fine-grained details and achieving a low compression rate.

In this work, we introduce **P**oint cloud **I**mplicit neural **CO**mpression (`PICO`), a framework that reformulates PCC from a signal processing problem to a neural network compression problem. `PICO` begins by modeling the point cloud's geometry and attributes using two compact INRs separately. Subsequently, `PICO` directly compresses the parameters of these learned INRs through advanced quantization and entropy coding techniques. This paradigm provides two key advantages. First, it decouples geometry and attribute modeling, avoiding the detrimental feature entanglement. Second, it separates point cloud representation from compression, enabling fine-grained control over the compression rate and reconstruction quality.

`PICO` incorporates a multi-scale rate control mechanism that allows precise and dynamic bitrate allocation, providing a notable advantage over existing methods. For coarse-grained control, we select an optimal model architecture using a pre-computed Pareto frontier that profiles the trade-off between model size and bitrate. To achieve finer-grained adjustments, we then apply a tunable $\ell_1$ regularization during training to promote parameter sparsity. This sparsity facilitates compression of the trained model, allowing the final bitrate to be precisely determined by adjusting the quantization step size during entropy coding. Through jointly optimization of model size, sparsity, and quantization, `PICO` achieves precise bitstream control while preserving high compression quality.

`PICO` adopts the Kolmogorov-Arnold Network (KAN) (Liu et al., 2025) as its INR backbone instead of the typical multilayer perceptron (MLP). Inspired by KAN, we design a backbone called **Le**arnable **A**ctivation **F**unction **Net**work (`LeAFNet`). Compared to MLP, its learnable activation functions can better capture the high-frequency details in point clouds, while achieving comparable INR performance with fewer parameters, which is a critical factor for compression tasks. To further enhance performance, we adapt `LeAFNet` for PCC by adding positional encoding to improve spatial understanding and replacing B-spline functions with radial basis functions to increase model throughput. These modifications make `LeAFNet` a backbone specifically designed for PCC.

`PICO` improves practical deployability by optimizing sampling space and strategy, as well as introducing dynamic thresholding, which together substantially reduce computational overhead and memory footprint. In addition, we explore how to extend the `PICO` from static to dynamic point clouds, broadening its applicability to a wider range of point cloud types.

We evaluated `PICO` against MPEG standards and other PCC methods on the 8iVFB, KITTI (Geiger et al., 2013), and ScanNet (Dai et al., 2017a) datasets, and conducted ablation studies to validate the effectiveness of our design. On the 8iVFB dataset, `PICO` showed strong performance. It reduced BD-BR by 53.54% and improved BD-PSNR by 4.92 dB for geometry compression, and in the more challenging joint compression task, it achieved a 42.71% BD-BR reduction and a $2.70 \times 10^{-3}$ improvement in BD-PCQM. These results provide strong evidence of the efficiency of `PICO`.

Our main contributions can be summarized as follows:

❶ We propose `PICO`, an implicit neural PCC framework with precise rate control mechanism, which is further optimized for real-world deployment and applicable to a wide range of point cloud types.

❷ We propose `LeAFNet`, an INR backbone with learnable activation functions, which is lightweight and highly effective at fitting implicit functions, making it well-suited for PCC.

❸ We reformulate PCC as neural network compression and conduct extensive experiments to assess its potential as a foundation or component of next-generation PCC.

## 2 RELATED WORK

### 2.1 POINT CLOUD COMPRESSION

The MPEG 3D Graphics Coding Group has established the PCC standard, introducing two methods: G-PCC and V-PCC (Graziosi et al., 2020). G-PCC directly encodes geometry and attributes, using an octree with entropy coding for voxelized geometry and Trisoup meshes for low-bitrate surface approximation, while attributes are handled by transforms such as region-adaptive hierarchical transform (RAHT) (De Queiroz & Chou, 2016). V-PCC projects point clouds into 2D patches and attribute maps via planar parameterization, then packs them into video frames for High Efficiency Video Coding (HEVC) (Sullivan et al., 2012) compression, using geometry-color separation and motion-compensated prediction to maintain continuity.

Recent advances in deep learning have significantly improved PCC with autoencoder-based frameworks, which encode point clouds into compact latent representations and reconstruct them via learned decoders, using entropy models to optimize the rate–distortion trade-off. To address point cloud irregularity, two main architectures have emerged: voxel-based methods (Wang et al., 2021b;a; 2022), which apply hybrid 3D convolutions on volumetric representations, and point-wise methods (Huang & Liu, 2019; Sheng et al., 2021), inspired by PointNet (Qi et al., 2017), which operate directly on raw point coordinates without voxelization artifacts. Although these approaches often outperform traditional codecs in rate–distortion performance, they continue to struggle with generalization to unseen domains and scalability to large-scale scenes (Quach et al., 2022).

To overcome the limitations of encoder–decoder frameworks, recent research has explored a new paradigm for PCC based on INRs (Ruan et al., 2024a; Xue et al., 2024; Ruan et al., 2024b). These methods represent an entire point cloud as a continuous, coordinate-conditioned neural function, which mitigates the generalization limitations of prior learning-based approaches. However, these works do not investigate how to select optimal INR backbone model for target bitrates, nor do they address efficiency considerations for practical deployment.

## 2.2 IMPLICIT NEURAL REPRESENTATION

INR (Ramasinghe & Lucey, 2022; Saragadam et al., 2023; Sitzmann et al., 2020) parameterize continuous multidimensional signals using coordinate-based neural networks. Given an input coordinate $\boldsymbol{x} \in \mathbb{R}^d$, a neural network $f_\theta$, whose parameters $\theta$ are optimized to minimize the reconstruction error with respect to the ground truth signal $s(\boldsymbol{x})$, outputs the corresponding signal value $f_\theta(\boldsymbol{x}) \approx s(\boldsymbol{x})$. Formally, an INR can be expressed as:

$$f_\theta : \boldsymbol{x} \in \mathbb{R}^d \mapsto s(\boldsymbol{x}) \in \mathbb{R}^c, \quad \theta^* = \arg\min_\theta \mathcal{L}\big(f_\theta(\boldsymbol{x}), s(\boldsymbol{x})\big), \tag{1}$$

where $\mathcal{L}$ denotes a suitable reconstruction loss, $d$ is the input coordinate dimension, and $c$ is the signal dimension. INRs have been widely used for data compression in other domains. For instance, COIN and other methods (Dupont et al., 2021; Strümpler et al., 2022; Dupont et al., 2022) map pixel coordinates to pixel colors and use meta-learning to improve fitting efficiency. These works offer valuable insights for developing INR-based PCC.

## 2.3 KOLMOGOROV-ARNOLD NETWORK

KAN (Liu et al., 2025) is a network architecture designed to improve function approximation and interpretability. Unlike MLP with fixed activation functions, KAN is inspired by the Kolmogorov–Arnold representation theorem, which states that any multivariate continuous function can be expressed as a superposition of univariate continuous functions. Leveraging this insight, KAN decomposes complex high-dimensional functions into combinations of simpler one-dimensional functions, using learnable one-dimensional functions as activation units. This design enables KAN to achieve exceptional representational efficiency, making it particularly effective for modeling high-frequency details and complex signal, and providing a powerful tool for INR-based applications.

## 3 PICO

In this section, we introduce PICO. We first briefly describe how the two-stage compression is implemented, and then present the improvements PICO makes for the two-stage process. Next, we describe how PICO achieves rate control and the design details of LeAFNet. The pseudocode of the compression and decompression algorithm is provided in detail in Alg. 1 and Alg. 2.

### 3.1 TWO STAGE COMPRESSION

The 3D point cloud $\mathcal{P} = \{\mathcal{X}, \mathcal{A}\}$ in $N$-bit voxelized space $\mathcal{S}$ typically comprises two components, geometry $\mathcal{X}$ representing a set of 3D coordinates, where the coordinate $\boldsymbol{x}$ satisfies:

$$\boldsymbol{x} = (x, y, z) \in \mathcal{S} = \left\{ \left( \frac{k_x}{2^N}, \frac{k_y}{2^N}, \frac{k_z}{2^N} \right) \,\bigg|\, k_x, k_y, k_z \in \mathbb{Z}, \ 0 \leq k_x, k_y, k_z < 2^N \right\}, \tag{2}$$

and attributes $\mathcal{A}$ representing color, material, or reflectance. `PICO` compresses the point cloud $\mathcal{P}$ by applying geometry compression to $\mathcal{X}$ and attribute compression to $\mathcal{A}$.

**Stage 1: Geometry Compression.** We train the first INR $f_g$ to learn a continuous occupancy field. $f_g$ takes a coordinate from $\mathcal{S}$ as input and outputs the probability $p$ that this coordinate is occupied by the geometry $\mathcal{X}$. Subsequently, we binarize the continuous field using a threshold $\hat{\tau}$, marking each coordinate as either occupied or unoccupied to obtain reconstructed geometry $\hat{\mathcal{X}}$:

$$f_g : \boldsymbol{x} \in \mathcal{S} \mapsto p \in [0, 1], \tag{3}$$

$$\hat{\mathcal{X}} = \{\boldsymbol{x} \mid f_g(\boldsymbol{x}) > \tau, \boldsymbol{x} \in \mathcal{S}\}. \tag{4}$$

**Stage 2: Attribute Compression.** After obtaining $\hat{\mathcal{X}}$, we train a second INR $f_a$ to learn the corresponding attributes. $f_a$ takes a coordinate from $\hat{\mathcal{X}}$ as input and outputs the normalized attribute $\boldsymbol{c}$ (consider color as the default attribute). We use $\tilde{\mathcal{A}}$ as the training ground truth, which is obtained by mapping attributes from $\mathcal{A}$ to $\hat{\mathcal{X}}$ using a nearest neighbor principle. By traversing all coordinates in $\hat{\mathcal{X}}$ through $f_a$, we obtain the reconstructed attributes $\hat{\mathcal{A}}$:

$$f_a : \boldsymbol{x} \in \hat{\mathcal{X}} \mapsto \boldsymbol{c} \in [0, 1]^3, \tag{5}$$

$$\tilde{\mathcal{A}}(\hat{\boldsymbol{x}}_i) = \mathcal{A}\left(\arg\min_{\boldsymbol{x}_j \in \mathcal{X}} \|\hat{\boldsymbol{x}}_i - \boldsymbol{x}_j\|\right), \tag{6}$$

$$\hat{\mathcal{A}} = \{f_a(\boldsymbol{x}) \mid \boldsymbol{x} \in \hat{\mathcal{X}}\}. \tag{7}$$

We consider the INR set $\{f_g, f_a\}$ as a proxy for the compressed point cloud. It is only necessary to store and transmit $\{f_g, f_a\}$. During point cloud decompression, we can obtain the reconstructed point cloud $\hat{\mathcal{P}} = \{\hat{\mathcal{X}}, \hat{\mathcal{A}}\}$ by traversing the spatial coordinates through Eq. 4 and Eq. 7.

### 3.2 SAMPLING SPACE AND STRATEGY

**Sampling Space.** Due to the inherent sparsity of the original point cloud, the vast majority of voxels in the voxelized space $\mathcal{S}$ are empty, which makes training and inference on the whole $\mathcal{S}$ computationally prohibitive and difficult to optimize. To address this issue, we divide the original space $\mathcal{S}$ into $2^M \times 2^M \times 2^M$ coarse-grained cubes, and `PICO` processes only the set of non-empty cubes, denoted as $\mathcal{W}$, during both training and inference. The optimized sampling space $\mathcal{V}$ is thus defined as the union of all voxels within these non-empty cubes:

$$\mathcal{W} = \{\boldsymbol{w} \mid \boldsymbol{w} = \lfloor \boldsymbol{x} \cdot 2^M \rfloor / 2^M, \boldsymbol{x} \in \mathcal{X}\}, \tag{8}$$

$$\mathcal{V} = \{\boldsymbol{x} \mid \lfloor \boldsymbol{x} \cdot 2^M \rfloor / 2^M \in \mathcal{W}, \boldsymbol{x} \in \mathcal{S}\}. \tag{9}$$

**Sampling Strategy** Although redefining the sampling space from $\mathcal{S}$ to $\mathcal{V}$ eliminates a large number of empty voxels, non-empty voxels still constitute only a tiny fraction $\delta$ within $\mathcal{V}$. This severe class imbalance presents a huge challenge for training. To mitigate this problem, we use weighted sampling to control the proportion of positive labels $\alpha = 0.5$ in each training batch $\mathbf{x}$. Sampling $\mathbf{x}$ from both the non-empty voxels $\mathcal{X}$ and empty voxels $\mathcal{V} \backslash \mathcal{X}$ can be expressed as:

$$\mathbf{x} = \alpha \cdot U(\mathcal{X}) \oplus (1 - \alpha) \cdot U(\mathcal{V} \backslash \mathcal{X}), \tag{10}$$

where $U(\cdot)$ denotes uniform sampling, $\oplus$ denotes the concatenation. However, computing the empty voxels $\mathcal{V} - \mathcal{X}$ incurs a time complexity of $O(|\mathcal{V}| \cdot |\mathcal{X}|)$, while storing them requires $O(|\mathcal{V}|)$ memory.

To reduce the time overhead, we avoid the costly operation of explicitly generating $\mathcal{V} \backslash \mathcal{X}$. Instead, We perform approximate sampling separately from the non-empty voxels $\mathcal{X}$ and the redefined sampling space $\mathcal{V}$. Eq. 10 can then be rewritten as:

$$\mathbf{x} = \hat{\alpha} \cdot U(\mathcal{X}) \oplus (1 - \hat{\alpha}) \cdot U(\mathcal{V}), \tag{11}$$

where the calibrated sampling rate $\hat{\alpha} = (\alpha - \delta)/(1 - \delta)$ is used to maintain the target class ratio. This strategy lowers the time complexity to $O(1)$, resulting in a more efficient sampling process.

---

**Algorithm 1** `PICO_Compression`

---

**Input**: point cloud $\mathcal{P} = \{\mathcal{X}, \mathcal{A}\}$, model dictionary $\mathcal{M}$, bitrate $b$, voxel space $\mathcal{S}$
**Parameter**: learning rate $\gamma$, cube resolution $M$, voxel resolution $N$
**Output**: bitstream $\widetilde{\theta}_g, \widetilde{\theta}_a$

1: $\mathcal{W} \leftarrow \{\boldsymbol{w} \mid \boldsymbol{w} = \lfloor \boldsymbol{x} \cdot 2^M \rfloor / 2^M, \boldsymbol{x} \in \mathcal{X}\}$
2: $\mathcal{V} \leftarrow \{\boldsymbol{x} \mid \lfloor \boldsymbol{x} \cdot 2^M \rfloor / 2^M \in \mathcal{W}, \boldsymbol{x} \in \mathcal{S}\}$
3: $\theta_g^{(0)}, \theta_a^{(0)}, \lambda_g, \lambda_a, \Delta_g, \Delta_a, T_g, T_a \leftarrow \mathcal{M}(b)$
4: **for** $t = 1$ **to** $T_g$ **do**
5: $\quad \mathbf{x} \leftarrow \texttt{Sample}(\mathcal{X}, \mathcal{V})$
6: $\quad \theta_g^{(t+1)} \leftarrow \theta_g^{(t)} - \gamma_g \nabla \mathcal{L}_{\text{geometry}}(f_g(\mathbf{x}; \theta_g^{(t)}, \lambda_g))$
7: **end for**
8: $\hat{\theta}_g \leftarrow \texttt{Quantization}(\theta_g^{(T_g)}, \Delta_g)$
9: $\mathcal{O} \leftarrow \{p \mid f_g(\boldsymbol{x}; \hat{\theta}_g), \boldsymbol{x} \in \mathcal{V}\}$
10: $\tau \leftarrow \texttt{AdaptiveThreshold}(\mathcal{O})$
11: $\hat{\mathcal{X}} \leftarrow \{\boldsymbol{x} \mid f_g(\boldsymbol{x}; \hat{\theta}_g) > \tau, \boldsymbol{x} \in \mathcal{V}\}$
12: **for** $t = 0$ **to** $T_a$ **do**
13: $\quad \mathbf{x} \leftarrow \texttt{Sample}(\hat{\mathcal{X}})$
14: $\quad \theta_a^{(t+1)} \leftarrow \theta_a^{(t)} - \gamma_a \nabla \mathcal{L}_{\text{attribute}}(f_a(\mathbf{x}; \theta_a^{(t)}, \lambda_a))$
15: **end for**
16: $\hat{\theta}_a \leftarrow \texttt{Quantization}(\theta_a^{(T_a)}, \Delta_a)$
17: $\widetilde{\theta}_g, \widetilde{\theta}_a \leftarrow \texttt{EntropyEncode}(\hat{\theta}_g, \hat{\theta}_a)$

---

Considering that we train our INRs on GPU and that a single-frame point cloud contains millions of points, the $O(|\mathcal{V}|)$ memory overhead is non-negligible. We split single coordinate $\boldsymbol{x}$ into two components, which are represented as:

$$\boldsymbol{x} = \boldsymbol{w} + \boldsymbol{w}' = \lfloor \boldsymbol{x} \cdot 2^M \rfloor / 2^M + \boldsymbol{w}', \tag{12}$$

$$\boldsymbol{w}' = (x, y, z) \in \mathcal{S}' = \left\{ \left( \frac{k_x}{2^N}, \frac{k_y}{2^N}, \frac{k_z}{2^N} \right) \,\middle|\, k_x, k_y, k_z \in \mathbb{Z}, \, 0 \leq k_x, k_y, k_z < 2^{N-M} \right\}. \tag{13}$$

where $\boldsymbol{w}$ denotes the coordinates of the non-empty cube containing $\boldsymbol{x}$, and $\boldsymbol{w}'$ represents the relative position of $\boldsymbol{x}$ within $\boldsymbol{w}$. Therefore, the process of sampling a batch of $\mathbf{x}$ can be expressed as:

$$\mathbf{x} = \mathbf{w} + \mathbf{w}' = \hat{\alpha} \cdot U(\mathcal{X}) \oplus (1 - \hat{\alpha}) \cdot (U(\mathcal{W}) + U(\mathcal{S}')). \tag{14}$$

By decomposing $\boldsymbol{x}$ and sampling separately from $\mathcal{W}$ and $\mathcal{S}'$, we avoid explicitly storing $\mathcal{V}$. In practice, it suffices to generate a set of random coordinates from $\mathcal{W}$ and $\mathcal{S}'$ independently to compute $\mathbf{x}$. This reduces the storage overhead to $O(|\mathcal{W}|)$. Considering that $O(|\mathcal{W}|)$ typically corresponds to only 0.02% of $O(|\mathcal{V}|)$, this optimization is highly significant.

### 3.3 DYNAMIC THRESHOLD

In Eq. 4, we need a threshold $\tau$ to classify points as either occupied or unoccupied. A static threshold is unlikely to perform robustly across diverse point clouds. So we introduce a dynamic thresholding mechanism to improve geometric reconstruction. We first define the set of all predicted occupancy probabilities within our sampling space $\mathcal{V}$ as $\mathcal{O}$:

$$\mathcal{O} = \{p \mid p = f_g(\boldsymbol{x}), \boldsymbol{x} \in \mathcal{V}\}. \tag{15}$$

We use D1 PSNR as the metric for geometry quality, denoted by $\mathcal{D}(\mathcal{O}, \tau)$. Empirically, we observed that this function is typically unimodal with respect to $\tau$ in Appendix B. Therefore, a golden section search algorithm can be employed to find the optimal $\tau$ that maximizes $\mathcal{D}(\mathcal{O}, \tau)$.

### 3.4 DYNAMIC RATE CONTROL

In this section, we describe how `PICO` achieves dynamic rate control from coarse-grained to fine-grained compression. The approach consists of three components: adaptive model selection, regularized training, and quantization with entropy coding.

**Algorithm 2** `PICO_DeCompression`

---

**Input**: bitstream $\widetilde{\theta}_g, \widetilde{\theta}_a$, space $\mathcal{V}$
**Parameter**: threshold $\tau$, model information $f_g, f_a$
**Output**: decompressed point cloud $\hat{\mathcal{P}} = \{\hat{\mathcal{X}}, \hat{\mathcal{A}}\}$

1: $\hat{\theta}_g, \hat{\theta}_a \leftarrow$ `EntropyDecode`$(\widetilde{\theta}_g, \widetilde{\theta}_a)$
2: $\hat{\mathcal{X}} \leftarrow \{\boldsymbol{x} \mid f_g(\boldsymbol{x}; \hat{\theta}_g) > \tau, \boldsymbol{x} \in \mathcal{V}\}$
3: $\hat{\mathcal{A}} \leftarrow \{f_a(\boldsymbol{x}; \hat{\theta}_a) \mid \boldsymbol{x} \in \hat{\mathcal{X}}\}$
4: $\hat{\mathcal{P}} \leftarrow \{\hat{\mathcal{X}}, \hat{\mathcal{A}}\}$

---

**Adaptive Model Selection.** Previous works often employ a single network architecture to compress point clouds of varying sizes. This approach can lead to significant performance degradation when the target bitrate changes. We observe that while small-parameter models struggle to compete with large-parameter models in compression quality at low compression rates, they gradually surpass larger models as the compression rate increases. This observation naturally led us to design a model dictionary for selecting the optimal model $f$ based on the target compression rate.

To construct this dictionary, we compress the same point cloud at different rates using models with various parameter counts. From their rate-distortion (RD) curves, we derive a Pareto frontier to define our model dictionary $\mathcal{M}$. In our experiments, we found that a model dictionary obtained from a single experiment can be broadly applied to a wide range of point clouds. For excessively large point clouds, we divide them into smaller blocks for processing. This ensures the high efficiency and practicality of our method for real-world deployment.

**Regularized Training.** We employ different objective functions for the two compression stages. For geometry compression, we use an $\alpha$-modulated focal loss Lin et al. (2017), which effectively addresses the class imbalance problem through a weighting mechanism. The balancing factor further corrects for the label imbalance between empty and occupied voxels caused by sampling bias. For attribute compression, we use a per-voxel mean squared error (MSE) loss, which is calculated only on occupied voxels to ensure perceptually accurate attributes. Furthermore, we add an $\ell_1$ regularization term during training to control the sparsity and distribution of the model parameters. The complete loss functions for geometry and attribute compression are as follows:

$$\mathcal{L}_{\text{geometry}} = \mathcal{L}_{\text{focal}} + \mathcal{L}_{\text{reg}} = \mathbb{E}_{p \sim \mathcal{O}}[-\alpha_t (1 - p_t)^\gamma \log(p_t)] + \lambda_g \|\Theta_g\|_1, \quad (16)$$

$$\mathcal{L}_{\text{attribute}} = \mathcal{L}_{\text{MSE}} + \mathcal{L}_{\text{reg}} = \mathbb{E}_{\boldsymbol{x} \sim \hat{\mathcal{X}}}[\|\tilde{\mathcal{A}}(\boldsymbol{x}_i) - f_a(\boldsymbol{x}_i)\|_2^2] + \lambda_a \|\Theta_a\|_1. \quad (17)$$

**Quantization & Entropy Coding.** Once the model parameters are obtained, they are quantized with step sizes $\Delta_g$ and $\Delta_a$ for geometry and attributes, respectively, and then entropy-coded to generate the final bitstream. Our framework utilizes DeepCABAC (Wiedemann et al., 2019), a specialized context-adaptive binary arithmetic coder designed for deep neural networks.

Suppose the model has $K$ parameters, denoted by $\theta$, with $\ell_1$ regularization strength $\lambda$ and a quantization step size $\Delta$. Then the average code length of the final bitstream of $\theta$ after entropy coding can be expressed as:

$$H(\theta) \approx K \log \frac{2e}{\lambda \Delta}. \quad (18)$$

We provide a derivation of this result in the Appendix C.

### 3.5 DYNAMIC POINT CLOUD COMPRESSION

We generalize the `PICO` framework to dynamic PCC, modeling the point cloud frame sequence as a function over a 4D spatio-temporal domain. By exploiting temporal redundancy, we augment the spatial coordinates $(x, y, z)$ with a time coordinate $t$. Consequently, our INRs become mappings which take a spatio-temporal coordinate $(x, y, z, t)$ as input and outputs a tuple containing the geometric occupancy $p \in \mathbb{R}$ or the attribute $\mathbf{c} \in \mathbb{R}^3$. The `PICO`-*Dynamic* can be expressed as:

$$f_g : (x, y, z, t) \mapsto p \in [0, 1], \quad \hat{\mathcal{X}}^{(t)} = \{\boldsymbol{x} \mid f_g(\boldsymbol{x}) > \tau_0^{(t)}, \boldsymbol{x} \in \mathcal{V}^{(t)}\}, \quad (19)$$

$$f_a : (x, y, z, t) \mapsto \mathbf{c} \in [0, 1]^3, \quad \hat{\mathcal{A}}^{(t)} = \{f_a(\boldsymbol{x}) \mid \boldsymbol{x} \in \hat{\mathcal{X}}^{(t)}\}. \quad (20)$$

### 3.6 LEARNABLE ACTIVATION FUNCTION NETWORK

We propose an INR backbone `LeAFNet` tailored for PCC in `PICO`. `LeAFNet` is designed to enhance the fitting of implicit functions for both geometric and attribute compression. The network structure is illustrated in Figure 1.

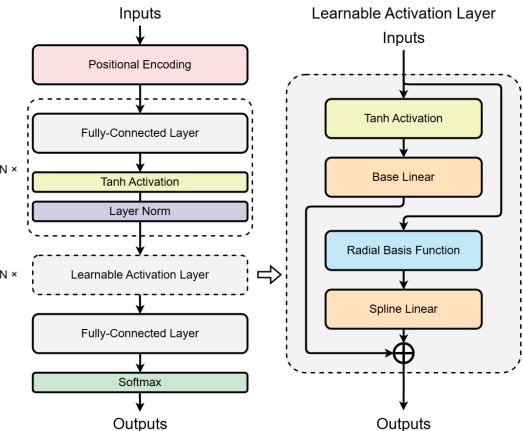

Figure 1: Model Architecture of `LeAFNet`.

**Positional Encoding.** `LeAFNet` processes 3D voxel coordinates through NeRF (Mildenhall et al., 2021)-style positional encoding to address the limited representational capacity of raw coordinate inputs. Specifically, each coordinate $\mathbf{x}$ is mapped to a higher-dimensional vector using a series of sinusoidal functions before being input to the network. This transformation allows the network to capture fine-grained geometric details and subtle attribute variations that would otherwise be lost in the raw representation, which is critical for high-fidelity reconstruction. The positional encoding $\Gamma$ can be expressed as:

$$\Gamma(\mathbf{x}; L) = (\mathbf{x}, \sin(2^0\pi\mathbf{x}), \cos(2^0\pi\mathbf{x}), \ ... \ , \sin(2^{L-1}\pi\mathbf{x}), \cos(2^{L-1}\pi\mathbf{x})). \qquad (21)$$

**Learnable Activation Layer.** The design of `LeAFNet` aims to integrate learnable activation functions to enhance implicit function approximation while maintaining parameter efficiency. The first part of `LeAFNet` is several fully connected layers, which reduce the dimensionality of the positional encoded input for subsequent processing. The core of LeafNet is the learnable activation function layer. Here, we adopt an approach similar to KAN, using the same structure as the `KAN_Layer`. The difference is that we replace the low-throughput B-spline functions with efficient radial basis functions (Li, 2024) (RBFs). The learnable activation function $\phi(x)$ is therefore defined as:

$$\phi(x) = w_b, \texttt{silu}(x) + w_s \sum_{i=1}^{N} \exp\left(-\frac{\|x - c_i\|^2}{h^2}\right), \qquad (22)$$

where $c_i$ and $h$ are hyperparameters that determine the shape of the activation function, and $w_b$ and $w_s$ are learnable linear layer weights. This design significantly accelerates both forward and backward passes, enabling faster computation and more stable training compared to KAN.

## 4 EXPERIMENTS

### 4.1 EXPERIMENTAL SETUPS

**Baselines.** We selected six state-of-the-art methods as our baselines for comparison: G-PCC (Graziosi et al., 2020), V-PCC (Graziosi et al., 2020), NeRC (Ruan et al., 2024b), NeRI (Xue et al., 2024), SparsePCGC (Wang et al., 2022), and Unicorn (Wang et al., 2024).

**Datasets** We conducted experiments on three widely recognized point cloud datasets: 8iVFB (d'Eon et al., 2019), KITTI (Geiger et al., 2012), and ScanNet (Dai et al., 2017b).

**Metrics.** In our experiments, we use the point-to-point error peak signal-to-noise ratio (D1 PSNR) (dB) to measure geometric distortion across all three datasets. For attribute distortion, we use dataset-specific metrics: **PCQM** (Meynet et al., 2020) ($\times 10^{-3}$) for the 8iVFB dataset, **R-PSNR** (dB) for KITTI, and **Y-PSNR** (dB) for ScanNet. To quantify the rate-distortion (RD) performance gains of different methods, we utilize the Bjontegaard delta metrics (Bjontegaard, 2001).

**Implementation Details.** We set the coarse-grained voxel resolution to $M = 5$, partitioning the space into $2^5 \times 2^5 \times 2^5$ cubes, each containing $32 \times 32 \times 32$ voxels. The network is optimized by Adam (Kingma & Ba, 2014), with an initial learning rate of $1 \times 10^{-3}$, which is decayed by a factor of 0.1 upon reaching a performance plateau. A batch size of 32,768 is used for training. For

| Method | KITTI | ScanNet | 8iVFB | | | | |
|---|---|---|---|---|---|---|---|
| Geo. Only | | | longdress | loot | redandblack | soldier | Average |
| G-PCC (octree) | -34.22 / 5.14 | -78.65 / 6.51 | -69.19 / 7.33 | -71.78 / 7.84 | -69.25 / 6.48 | -60.76 / 6.69 | -67.75 / 7.09 |
| G-PCC (trisoup) | -28.25 / 4.05 | -71.24 / 5.29 | -49.34 / 3.96 | -53.33 / 4.36 | -47.20 / 3.33 | -42.75 / 4.15 | -48.16 / 3.95 |
| V-PCC | -15.64 / 2.88 | -56.42 / 3.67 | -35.42 / 2.64 | -44.66 / 3.64 | -53.27 / 4.42 | -45.51 / 4.20 | -44.72 / 3.73 |
| NeRC | -18.57 / 3.65 | -58.23 / 4.05 | -33.13 / 2.80 | -45.53 / 4.56 | -48.12 / 4.92 | -43.55 / 4.36 | -42.58 / 4.16 |
| NeRI | -20.14 / 3.28 | -54.78 / 3.02 | -31.28 / 2.64 | -49.89 / 6.54 | -48.65 / 6.43 | -45.47 / 5.91 | -43.89 / 5.28 |
| SparsePCGC | -8.94 / 1.13 | -51.98 / 3.81 | -17.23 / 0.67 | -22.54 / 1.25 | -26.51 / 1.03 | -18.28 / 1.29 | -21.14 / 1.06 |
| Unicorn | 0.66 / -0.16 | 2.83 / -0.25 | 13.24 / -1.22 | 11.15 / -0.43 | 10.98 / -0.61 | 9.27 / -0.98 | 11.16 / -0.81 |

| Method | KITTI | ScanNet | 8iVFB | | | | |
|---|---|---|---|---|---|---|---|
| Geo. & Attr. | | | longdress | loot | redandblack | soldier | Average |
| G-PCC (octree) | -13.48 / 1.24 | -25.61 / 2.32 | -40.94 / 3.89 | -74.72 / 6.86 | -71.68 / 5.86 | -65.87 / 7.69 | -63.30 / 6.08 |
| G-PCC (trisoup) | -9.43 / 0.81 | -21.20 / 1.28 | -19.88 / 0.84 | -47.68 / 1.43 | -40.30 / 1.73 | -57.49 / 2.37 | -41.34 / 1.60 |
| V-PCC | -4.17 / 0.36 | -8.63 / 0.55 | -8.19 / 0.68 | -29.53 / 0.22 | -46.48 / 1.15 | -26.18 / 1.08 | -27.60 / 0.78 |
| NeRC | -5.20 / 0.54 | -7.09 / 0.39 | -17.19 / 1.35 | -49.41 / 2.51 | -51.77 / 2.51 | -43.68 / 3.06 | -40.51 / 2.36 |
| NeRI | -9.20 / 1.25 | -10.10 / 0.51 | -23.48 / 1.88 | -48.41 / 3.78 | -46.37 / 3.42 | -28.74 / 5.64 | -36.75 / 3.68 |
| SparsePCGC | -0.53 / 0.05 | -15.52 / 0.94 | -12.92 / 0.88 | -18.49 / 1.56 | -17.83 / 1.12 | -17.04 / 1.92 | -16.57 / 1.37 |
| Unicorn | 19.64 / 0.30 | 2.86 / -0.09 | 10.32 / -2.31 | 8.95 / 0.14 | 6.13 / -0.88 | 7.48 / -0.23 | 8.22 / -0.89 |

Table 1: We evaluated `PICO`'s compression performance against seven baselines on three datasets. The top part of the table shows the results for geometry-only compression, while the bottom part displays the results for joint geometry and attribute compression. We report the Bjontegaard delta (BD) gains of `PICO` relative to baseline methods. The number before the slash indicates the BD-Rate (%), where a lower value is better. The number after the slash represents the corresponding metric, where a higher value is better. RD curve visualization can be found in Appendix A.

geometry compression, we set the reweighted sampling coefficient to $\alpha = 0.5$ and the focal loss modulation coefficient to $\gamma = 2$. The quantized parameters from both models are then losslessly compressed using DeepCABAC (Wiedemann et al., 2019).

## 4.2 POINT CLOUD COMPRESSION

**Static Point Cloud.** To evaluate `PICO`'s performance, we selected a single frame from each of the four point cloud sequences in the 8iVFB dataset, and used individual point clouds from the KITTI and ScanNet datasets. The results are shown in Table 1. `PICO` achieves substantial improvements over conventional MPEG standards (G-PCC and V-PCC), delivering higher compression efficiency and reconstruction quality. It also surpasses existing INR-based approaches (NeRC and NeRI) and demonstrates a clear advantage over SparsePCGC. Nevertheless, there remains a small performance gap compared to the current state-of-the-art, Unicorn.

`PICO`'s improvements are especially notable in geometry compression, thanks to its continuous occupancy representation, which fits well with the nature of INRs. Attribute compression is more challenging, mainly because geometric errors accumulate and sharp attribute changes are difficult to capture with the smooth mappings of INR.

| Method | 8iVFB_longdress | |
|---|---|---|
| Geo. Only | BD-BR (%) | BD-PSNR (dB) |
| V-PCC | -59.51 | 5.30 |
| NeRC | -46.25 | 4.38 |
| Unicorn | -3.68 | 0.24 |

| Method | 8iVFB_longdress | |
|---|---|---|
| Geo. & Attr. | BD-BR (%) | BD-PCQM ($\times 10^{-3}$) |
| V-PCC | -23.98 | 1.25 |
| NeRC | -26.81 | 2.14 |
| Unicorn | 6.22 | -0.51 |

Table 2: `PICO` vs. baselines on the first 30 frames of `8iVFB_longdress`. Top: geometry compression. Bottom: joint compression.

**Dynamic Point Cloud.** To evaluate `PICO`-*dynamic*'s performance, we selected the first 30 frames of the `8iVFB_longdress` sequence. We benchmarked `PICO` against state-of-the-art methods, including V-PCC, NeRC, and Unicorn. The experimental results in Table 2 clearly demonstrate that `PICO` exhibits superior compression performance on dynamic data. `PICO` consistently outperforms both V-PCC and NeRC. A noteworthy finding is that `PICO` successfully surpasses Unicorn in geometry compression, highlighting the strength of our spatio-temporal geometry representation.

### 4.3 IMPACT OF LeAFNet

| Parameters | | | 8iVFB_loot | |
|---|---|---|---|---|
| $d$ | $L$ | $\alpha$ | BD-BR (%) | BD-PSNR (dB) |
| 24 | 64 | 0.5 | 25.15 | 0.61 |
| 48 | 64 | 0.5 | -11.16 | -0.11 |
| 36 | 48 | 0.5 | -5.36 | 0.89 |
| 36 | 32 | 0.5 | -25.64 | 1.36 |
| 36 | 16 | 0.5 | -34.21 | 2.68 |
| 36 | 0 | 0.5 | -41.63 | 3.38 |
| 36 | 64 | 0.25 | -39.09 | 2.93 |
| 36 | 64 | $\delta$ | -61.78 | 5.63 |

Table 3: Ablation study of LeAFNet's hyperparameters. Default $\{d, L, \alpha\} = \{36, 64, 0.5\}$

We conducted a comprehensive ablation study to investigate the impact of LeAFNet's hyperparameters—hidden layer dimension $d$, positional encoding dimension $L$, and sampling ratio $\alpha$—on compression performance. For this analysis, we established a default configuration with $d = 36$, $L = 64$, and $\alpha = 0.5$, and then varied each hyperparameter to evaluate its individual and combined effects on PICO. The results are summarized in Table 3.

Our analysis indicates that both the positional encoding and the sampling strategy are crucial for enhancing LeAFNet's performance. Increasing the positional encoding dimension $L$ improves compression by enabling the model to capture finer spatial details. Likewise, adjusting the positive label ratio $\alpha$ from its original, highly imbalanced value $\delta$ to 0.5 leads to a substantial gain in compression quality, as this resampling strategy mitigates training instability and facilitates learning a more robust occupancy probability distribution. The hidden layer dimension $d$ also plays an important role: larger values of $d$ increase the bitrate but simultaneously improve performance, highlighting the effective scalability of the LeAFNet architecture.

#### 4.3.1 ADAPTIVE MODEL PARAMETER SELECTION

To provide a more intuitive understanding of the model dictionary $\mathcal{M}$, we constructed a model dictionary on the 8iVFB_longdress, as illustrated in Figure 2. We varied the hidden layer dimensions $d \in 16, 32, 48, 64, 80, 96$ and network depths (number of layers) $\in 2, 3, 4, 5, 6, 7$, and assigned different training hyperparameters to each configuration to achieve a balance between performance and compression efficiency. Our experiments confirm the trends observed in earlier analyses: at lower bitrates, smaller models tend to perform better, as excessive quantization steps $\Delta$ and strong $\ell_1$ regularization $\lambda$ can degrade the representational capacity of larger models; conversely, at higher bitrates, larger models dominate, with their increased parameter capacity allowing them to better capture fine-grained point cloud details.

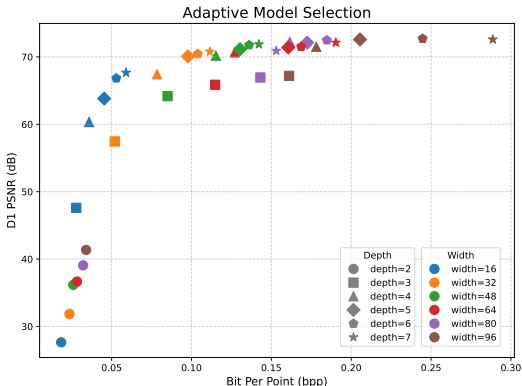

Figure 2: The 8iVFB_longdress model dictionary, also applicable to other point clouds.

## 5 CONCLUSION

In this work, we introduced PICO, a INR-based PCC framework that decouples geometry and attribute compression. By reformulating PCC as a neural network compression, PICO achieves flexible control over bitrate and reconstruction quality. We further proposed LeAFNet, a lightweight INR backbone with learnable activation functions, positional encoding and radial basis functions, which effectively capture high-frequency point cloud details with fewer parameters.

Extensive experiments on 8iVFB, KITTI, and ScanNet datasets demonstrate that PICO consistently outperforms traditional MPEG standards and existing PCC methods, achieving substantial gains in both geometry and joint compression metrics. Overall, PICO represents a significant step toward next-generation point cloud compression, providing a flexible, high-performance, and deployable solution that bridges INR and practical compression needs.

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

## A  RD CURVE VISUALIZATION

In Figure 3, we visualize the RD curves on three static point clouds, including the five methods: PICO (Ours), G-PCC (Graziosi et al., 2020), NeRC (Ruan et al., 2024b), SparsePCGC (Wang et al., 2022), and Unicorn (Wang et al., 2024). It can be found that PICO is basically on par with the state-of-the-art Unicorn, showing a significant performance improvement compared to the other methods.

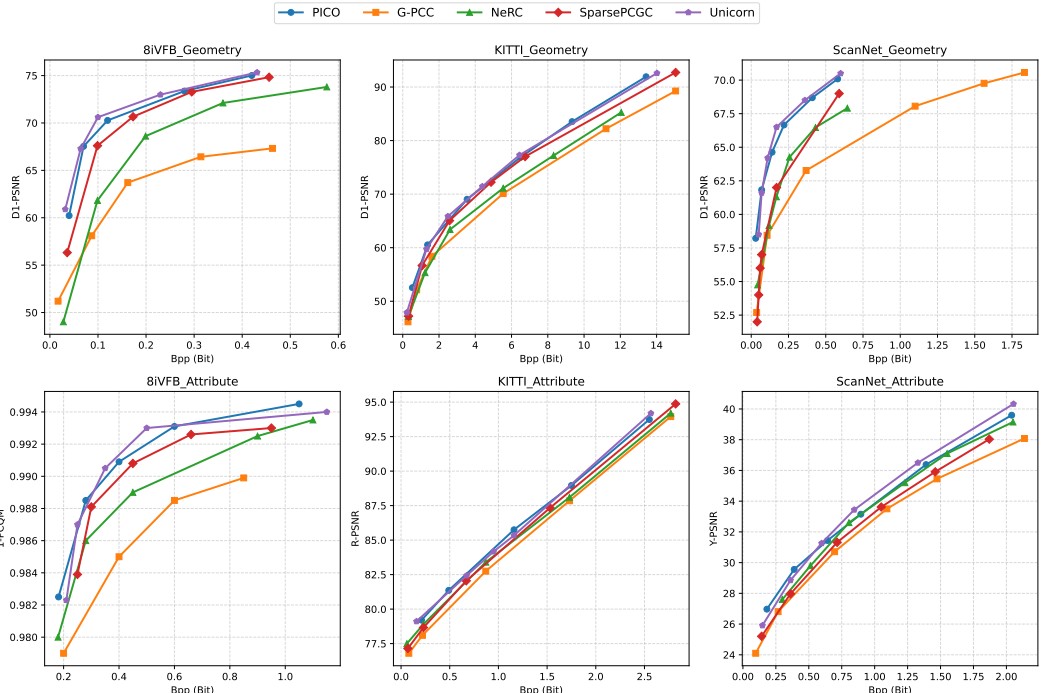

Figure 3: RD Curves of five methods on three different point clouds.

## B  UNIMODALITY OF $D(\mathcal{O}, \tau)$

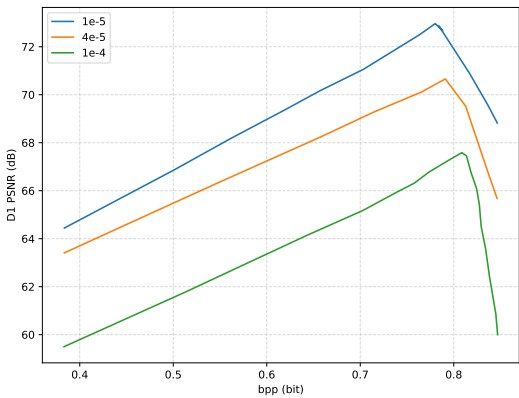

Figure 4: Unimodality of $D(\mathcal{O}, \tau)$

In Figure 4, we visualize a set of $D(\mathcal{O}, \tau)$ curves. For the same model, the D curves under different regularization strengths all exhibit unimodality with respect to $\tau$. Moreover, it can be observed that by adjusting $\tau$, we can achieve up to an 8 dB gain, demonstrating the effectiveness of our dynamic thresholding method.

## C   Derivation of $H(\theta) \approx K \log \frac{2e}{\lambda\Delta}$

**Theorem 1** (Average coding length of quantized $\ell_1$-regularized parameters). *Suppose a model has $K$ parameters $\theta = (\theta_1, \ldots, \theta_K)$, each subject to $\ell_1$ regularization with strength $\lambda$, and the parameters are quantized with step size $\Delta$. Then the average code length of the final bitstream of $\theta$ after entropy coding is approximately*

$$H(\theta) \approx K \log \frac{2e}{\lambda\Delta}.$$

*Proof.* We provide a detailed derivation of Eq.18 as follows.

Consider a single model parameter $\theta_i$ under $\ell_1$ regularization. Its prior distribution is the Laplace distribution, with probability density function (PDF):

$$p(\theta_i \mid \lambda) = \frac{\lambda}{2} \exp(-\lambda|\theta_i|),$$

where $\lambda$ is the regularization strength. This distribution forms the basis of our entropy calculation.

The differential entropy of a continuous random variable $X$ is defined as

$$H(X) = -\int_{-\infty}^{\infty} p(x) \log p(x) \, dx.$$

Substituting the Laplace PDF, we have

$$
\begin{aligned}
H(\theta_i) &= -\int_{-\infty}^{\infty} \frac{\lambda}{2} e^{-\lambda|\theta_i|} \log\left(\frac{\lambda}{2} e^{-\lambda|\theta_i|}\right) d\theta_i \\
&= -\int_{-\infty}^{\infty} \frac{\lambda}{2} e^{-\lambda|\theta_i|} \left[\log\left(\frac{\lambda}{2}\right) - \lambda|\theta_i|\right] d\theta_i \\
&= -\left[\log\left(\frac{\lambda}{2}\right) \int_{-\infty}^{\infty} \frac{\lambda}{2} e^{-\lambda|\theta_i|} d\theta_i - \lambda \int_{-\infty}^{\infty} |\theta_i| \frac{\lambda}{2} e^{-\lambda|\theta_i|} d\theta_i\right].
\end{aligned}
$$

The first integral evaluates to 1, because it is the integral of the PDF over the whole real line. The second integral is the expected value of $|\theta_i|$ under the Laplace distribution, which is $E[|\theta_i|] = 1/\lambda$. Therefore, we obtain

$$H(\theta_i) = -\left[\log\left(\frac{\lambda}{2}\right) - \lambda \cdot \frac{1}{\lambda}\right] = \log\left(\frac{2e}{\lambda}\right),$$

which is the differential entropy of a single parameter.

If we quantize $\theta_i$ with a small step size $\Delta$, the discrete entropy of the quantized variable, $H_\Delta(\theta_i)$, is approximately related to the differential entropy by

$$H_\Delta(\theta_i) \approx H(\theta_i) - \log\Delta = \log\left(\frac{2e}{\lambda\Delta}\right).$$

Considering the $K$ parameters $\theta_1, \ldots, \theta_K$ are drawn from the same distribution $\theta$ obtained from a single training run, the total entropy of the parameter set is the sum of the individual entropies:

$$H(\theta) = \sum_{i=1}^{K} H_\Delta(\theta_i) \approx \sum_{i=1}^{K} \log\left(\frac{2e}{\lambda\Delta}\right) = K \log\left(\frac{2e}{\lambda\Delta}\right).$$

This final expression approximates the total number of nats required to encode all quantized parameters optimally. $\square$

## D BASELINES & DATASETS

We selected six methods as our baselines for comparison, representing a diverse range of point cloud compression techniques.

- **G-PCC** (Graziosi et al., 2020) is a geometry-based PCC standard that uses an octree structure and entropy coding to compress voxelized geometry and attributes.
- **V-PCC** (Graziosi et al., 2020) is a video-based PCC standard that projects 3D point clouds onto 2D planes for compression using video codecs.
- **NeRC** (Ruan et al., 2024b) is a PCC framework that uses two separate neural networks to implicitly represent point cloud's geometry and attributes.
- **NeRI** (Xue et al., 2024) compresses point clouds by projecting 3D frames into 2D range images and encoding them via an implicit neural network.
- **SparsePCGC** (Wang et al., 2022) is a multiscale sparse tensor point cloud geometry compression method.
- **Unicorn** (Wang et al., 2024) is a versatile, multiscale conditional coding framework that uses spatial and temporal scale priors to jointly compress point clouds.

We conducted experiments on three widely recognized point cloud datasets to demonstrate the versatility of our method and network.

- **8iVFB** (d'Eon et al., 2019) is a dynamic voxelized point cloud dataset containing sequences of human subjects captured at high resolution, primarily used for evaluating point cloud compression standards like MPEG.
- **KITTI** (Geiger et al., 2012) is a popular dataset for autonomous driving research, featuring a rich collection of outdoor urban scenes with synchronized data from multiple sensors, including a 3D LiDAR scanner, stereo cameras, and GPS/IMU.
- **ScanNet** (Dai et al., 2017b) is a large-scale RGB-D video dataset of indoor scenes, providing richly annotated 3D reconstructions with instance-level semantic segmentations, used for various 3D scene understanding tasks.

## E GENERATIVE AI USAGE STATEMENT

In the preparation of this document, We have used generative AI tools only for grammar checking and language refinement. All content, ideas, and technical material were developed independently, and the AI was not used to generate original content, perform substantive analysis, or contribute to the intellectual work itself.

