# OpenReview forum: "Implicit Neural Compression of Point Clouds via Learnable Activation Function"
_ICLR.cc/2026/Conference — Submitted to ICLR 2026_

### Official Review · Reviewer_Fun1 · 2025-10-28

**Soundness:** 3
**Presentation:** 3
**Contribution:** 2
**Rating:** 4
**Confidence:** 3

**Summary:**

This paper introduces Point cloud Implicit neural COmpression (PICO), a framework that transforms point cloud compression from a signal processing issue to a neural network compression problem by modeling geometry and attributes separately with compact implicit neural representations and compressing their parameters via quantization and entropy coding. This approach decouples geometry and attribute modeling to prevent feature entanglement and separates representation from compression for precise control over rate and quality. PICO employs a multi-scale rate control mechanism, using a pre-computed Pareto frontier for coarse-grained architecture selection and tunable L1 regularization for finer-grained parameter sparsity, enabling precise bitrate allocation through quantization step size adjustment. The framework adopts the Kolmogorov-Arnold Network-inspired Learnable Activation Function Network as its INR backbone, which captures high-frequency details with fewer parameters and is further enhanced for PCC with positional encoding and radial basis functions. Evaluations on the 8iVFB, KITTI, and ScanNet datasets demonstrate its superiority over MPEG standards and other PCC methods. The contributions include PICO's precise rate control and real-world optimization, the lightweight and effective LeAFNet backbone, and the reformulation of PCC as neural network compression.

**Strengths:**

I am fond of the proposed idea.  While I cannot confirm whether an identical approach has been previously applied to point cloud compression, it is worth noting that similar strategies have proven highly effective in the fields of 3D representation and 3D reconstruction/novel view synthesis.  The paper is well-articulated, with most concepts clearly explained.  Furthermore, the simplicity and directness of the algorithm's implementation stand out as a significant advantage.  In the experimental section, the paper provides a relatively thorough analysis of Adaptive Model Parameter selection.

**Weaknesses:**

My concerns regarding this work are as follows:

1) As indicated by the quantitative comparison experiments, the proposed method does not surpass, and in some cases, slightly underperforms compared to the previous state-of-the-art method, Unicorn. This diminishes my confidence in the method's overall effectiveness.
2) What is the speed performance of the proposed method? The authors should report the compression speed and provide a comparative analysis with other methods. Slow processing speed or low efficiency is widely recognized as a drawback of INR-based approaches, and I am curious to know whether the proposed method suffers from this limitation.
3) The network architecture employed in the paper lacks innovation, as it primarily consists of several generic modules, including Positional Encoding, Quantization & Entropy Coding, and Regularized Training.

**Questions:**

Please refer to the questions raised in the Weaknesses section. I hope the authors can provide responses to these issues in the rebuttal.

---

> ### Author Response · Authors · 2025-11-12
> **Response to Fun1**
>
> We thank you for the reviewer Fun1’s valuable comments. Our detailed responses to each weakness are provided below.
>
> 1. Our method can approach the performance of Unicorn very closely (In fact, PICO surpasses Unicorn in dynamic point cloud geometry compression. Please refer to Table 1, Table 2, and Figure 3). And we believe that PICO still holds great potential:
>
>    - **Scalability:** Rather than being a specific method, PICO can be viewed as a *framework* for point cloud compression that differs fundamentally from codec-based approaches. It reformulates the point cloud compression problem into a model compression problem, allowing it to incorporate advanced techniques from the ML systems community. For example, in our paper, we only use simple quantization and entropy coding to compress the model parameter. In practice, we can further adopt more advanced quantization methods and introduce quantization-aware training during the learning process. We hope PICO can inspire the multimedia community to explore alternatives to codec-based methods and ultimately provide more comprehensive point cloud compression technologies in the future.
>
>    - **Stability and Controllability:** Methods like Unicorn rely on entropy control of latent features during training, which can lead to overfitting, training collapse, and a lack of explicit bitrate control mechanisms. In Section 3.4 and Appendix C, we provide a detailed analysis of how controlling the model size can effectively regulate the compression bitrate. This approach avoids the “black-box” nature of prior compression mechanisms—something previous methods have not achieved.
>
>    - **Applicability to large-scale or rare point clouds with limited training data:** For such point clouds, codec-based methods often generalize poorly and thus perform inefficiently. Our method, however, does not suffer from this issue, as it can be trained directly on a single point cloud and supports cube-wise compression.
>
>
> 2. There are still issues with training speed in methods based on INR. However, considering that our task is compression and the backbone model has a small number of parameters, most models are kept around 10^5 parameters. We conducted an encoding and decoding experiment to study the average speed of four point cloud geometries from 8iVFB, measuring the time required to compress each frame (s/frame).
>
>     | s/frame | G-PCC | V-PCC | Unicorn | PICO | PICO-dynamic |
>     | ------- | ----- | ----- | ------- | ---- | ------------ |
>     | Encode  | 2.71  | 25.9  | 89.3    | 1059 | 18.6         |
>     | Decode  | 0.9   | 3.2   | 67.2    | 5.8  | 5.6          |
>
>     PICO-dynamic performs compression over 60-frame dynamic point clouds. The following conclusions can be drawn:
>
>     * **PICO** is relatively slow in compression speed (as it trains on a single point cloud), but it has an advantage in decompression speed.
>     * **PICO-dynamic**, when compressing 60 frames simultaneously, can still maintain a fast per-frame compression speed while achieving high compression quality (as shown in Table 2, its dynamic point cloud geometry compression quality surpasses Unicorn). Therefore, in practical applications, multiple point cloud frames can be compressed simultaneously to maintain compression efficiency.
>
> 3. Our main innovations are as follows:
>    * We transform the point cloud compression problem into a model compression problem, enabling the application of efficient machine learning methods to computer vision tasks.
>    * We propose an INR-based compression framework called PICO, design a high-precision bitrate control mechanism, and analyze how hyperparameters affect compression rate and quality.
>    * We introduce KAN into INR-based compression and design a new backbone network called LeAFNet.
>
>    In terms of implementation, although we make extensive use of methods that have already been proven effective, we believe this does not imply a lack of innovation. Instead, by integrating quantization, entropy coding, and regularized training with the point cloud compression framework and bitrate control, we demonstrate that our method is highly extensible and effective in framework design. Even with the simplest implementations, it can achieve significant performance improvements.
>
> We believe these responses address the reviewers’ concerns. Please feel free to let us know if there are any further questions.

---

### Official Review · Reviewer_Ydhe · 2025-10-29

**Soundness:** 2
**Presentation:** 3
**Contribution:** 2
**Rating:** 4
**Confidence:** 4

**Summary:**

This paper proposes a point cloud compression method based on implicit neural representations. It first converts the point cloud into a voxel representation, and then trains a neural network to predict the occupancy and attributes of each point. Finally, the point cloud is compressed by coding the parameters of the trained network.

**Strengths:**

The paper is clearly structured and easy to follow. The proposed method sounds reasonable, and the quantitative results are also good.

**Weaknesses:**

1. Is the INR trained individually for each point cloud? Or can a single trained network generalize to arbitrary point clouds? If the model needs to be optimized separately for every point cloud, what is the required training time? In that case, its efficiency would be far inferior to feed-forward based methods.

2. In Line 264, it is mentioned that the optimal 𝑡 is searched to maximize the D1 PSNR. Is this search process performed only for the proposed method, or for all baselines as well? If only applied to the proposed method, the comparison would not be fair.

3. The Sampling Strategy section introduces several tricks to reduce training time and memory consumption. I would like to know how much practical improvement they bring — e.g., how many minutes of training are saved, and how much memory is reduced?

4. Why does the paper only provide quantitative results but no visualization results?

5. Prior works [1, 2] typically evaluate the decompressed point clouds on downstream tasks (e.g., object detection on KITTI) to demonstrate effectiveness. This paper lacks such experiments.

6. The paper does not include comparisons with some key baselines [3, 4], even though these baselines provide open-source implementations.

7. Regarding the backbone, how significant is the performance difference between using a KAN-based architecture and using an MLP?

[1] Que, Zizheng, Guo Lu, and Dong Xu. "Voxelcontext-net: An octree based framework for point cloud compression." CVPR, 2021.

[2] Huang, Lila, et al. "Octsqueeze: Octree-structured entropy model for lidar compression." CVPR, 2020.

[3] Fu, Chunyang, et al. "Octattention: Octree-based large-scale contexts model for point cloud compression." AAAI, 2022.

[4] You, Kang, et al. "Reno: Real-time neural compression for 3d lidar point clouds." CVPR, 2025.

**Questions:**

Please refer to the Weaknesses section for details.

---

> ### Author Response · Authors · 2025-11-13
> **Response to Ydhe (Part1)**
>
> We thank you for the reviewer Ydhe's valuable comments. Our detailed responses to each weakness are provided below.
>
> 1. The INR is trained separately for each point cloud. We conducted an encoding and decoding experiment to study the average speed of four point cloud geometries from 8iVFB, measuring the time required to compress each frame (s/frame).
>
>     | s/frame | G-PCC | V-PCC | Unicorn | PICO | PICO-dynamic |
>     | ------- | ----- | ----- | ------- | ---- | ------------ |
>     | Encode  | 2.71  | 25.9  | 89.3    | 1059 | 18.6         |
>     | Decode  | 0.9   | 3.2   | 67.2    | 5.8  | 5.6          |
>
>     PICO-dynamic performs compression over 60-frame dynamic point clouds. The following conclusions can be drawn:
>
>     * **PICO** is relatively slow in compression speed (as it trains on a single point cloud), but it has an advantage in decompression speed.
>     * **PICO-dynamic**, when compressing 60 frames simultaneously, can still maintain a fast per-frame compression speed while achieving high compression quality (as shown in Table 2, its dynamic point cloud geometry compression quality surpasses Unicorn). Therefore, in practical applications, multiple point cloud frames can be compressed simultaneously to maintain compression efficiency.
> 2. In the baselines of Table 1, the dynamic threshold $\tau$ is only applicable to NeRC and has already been applied. For other codec-based baseline methods, this step is not included. This is because our method models the voxel space as a continuous probability field, and searching for the optimal threshold $\tau$ can effectively reduce surface artifacts and improve the perceptual quality of the reconstructed point cloud.
>
> 3. We take longdress from the 8iVFB dataset as an example, which contains approximately $10^6$ points. I will explain the efficiency improvements in terms of both space and time:
>
>    - **Space Efficiency Optimization:** According to the unoptimized sampling method in Equation 10, we need to store $10^6 + 2^{30} (|\mathcal{X}| + 2^{3N})$ points on the GPU, representing non-empty voxels and empty voxels, respectively. However, when we switch to the sampling method in Equation 14, we only need to store $10^6 + 10^3(|\mathcal{X}| + |\mathcal{W}|)$ points, representing non-empty voxels and non-empty cubes. In other words, this requires only about **1/1000** of the original GPU memory. This reduction is crucial for compressing multiple point clouds simultaneously and for compressing dynamic point clouds.
>
>    - **Time Efficiency Optimization:** With the unoptimized sampling method in Equation 10, we need to compute $\mathcal{V - X}$ when initializing the sampling process, which takes roughly 10 seconds. After improving the method to the sampling strategy in Equation 14, we can skip this step entirely and simply generate random indices to perform sampling efficiently.
>
>
> 4. Thank you for your comment. We will update this section in the revised PDF after collecting all reviewers’ feedback.

---

> ### Author Response · Authors · 2025-11-13
> **Respond to Ydhe (Part II)**
>
> 5. Thank you for this comment. Following the experiment in Section 4.2.4 of Reno, we evaluated our method and Reno on the KITTI dataset. Here, we report the object detection performance on the *Car*, *Pedestrian*, and *Cyclist* point clouds. Since we cannot include figures, we report BD-mAP. We will update this part of the results in the revised PDF.
>     | BD-mAP | Car  | Pedestrian | Cyclist | Average |
>     |-------|------------|----------|------------|----------|
>     | Reno  | 2.63 | 3.51 | 2.96 | 3.03 |
>
>     It can be seen that the metrics we use to evaluate point cloud quality do indeed reflect the quality of point clouds when applied to downstream tasks.
>
> 6. Thank you for your suggestion. We conducted geometry compression evaluations of these methods on the four point clouds of the 8iVFB dataset.
>     | BD-BR/BD-PSNR | longdress | loot | redandblack | soldier | Average |
>     |-------|------------|----------|--------------|----------|----------|
>     | Octattention  | -31.57 / 1.65 | -34.52 / 3.35 | -39.37 / 1.58 | -35.41 / 2.96 | -35.21 / 2.38 |
>     | Reno          | -23.25 / 1.35 | -15.86 / 0.36 | -19.89 / 0.68 | -17.38 / 1.24 | -19.09 / 0.91 |
> 7. The baseline NeRC used in Table 1 also employs MLP as the backbone. Our method achieves significant improvements over it. The improvement comes not only from backbone differences but also from training details. To demonstrate this, we conducted ablation studies on geometry compression of 8iVFB, as shown below. It can be observed that LeAFNet and the model dictionary bring significant improvements to our method.
>    - NeRC
>    - PICO-MLP (LeAFNet replaced with MLP, w/o the model dictionary)
>    - PICO-MLP-dict (LeAFNet replaced with MLP, w/ the model dictionary)
>
>     | BD-BR/BD-PSNR | longdress | loot | redandblack | soldier | Average |
>     |-------|------------|------|--------------|----------|----------|
>     | NeRC | -33.13 / 2.80 | -45.53 / 4.56 | -48.12 / 4.92 | -43.55 / 4.36 | -42.58 / 4.16 |
>     | PICO-MLP | -19.37 / 1.54 | -27.97 / 3.03 | -26.37 / 2.52 | -23.11 / 2.65 | -24.20 / 2.43 |
>     | PICO-MLP-dict | -16.54 / 0.72 | -23.87 / 1.63 | -24.84 / 1.07 | -19.24 / 1.47 | -16.12 / 1.22 |
>
>
>
> We believe these responses address the reviewers’ concerns. Please feel free to let us know if there are any further questions.

---

### Official Review · Reviewer_mYum · 2025-10-31

**Soundness:** 2
**Presentation:** 3
**Contribution:** 2
**Rating:** 4
**Confidence:** 5

**Summary:**

In this paper, the authors present a point cloud compression framework that employs a Kolmogorov–Arnold Network (KAN) as an alternative to traditional backbone architectures. To enhance computational efficiency, the method partitions the 3D space into subspaces and processes only the occupied voxel blocks. The framework enables variable-rate compression by using models from a predefined model dictionary with varying parameter counts. Based on rate–distortion (RD) curve analysis, the most suitable model is selected for a given compression rate. Geometry and attribute data are compressed separately, with attribute compression dependent on the reconstructed geometry. Additionally, the framework adopts dynamic thresholding to determine voxel occupancy. The authors evaluate their method across three benchmark datasets, demonstrating its effectiveness for efficient, adaptive point cloud compression.

**Strengths:**

1. The use of the Kolmogorov–Arnold Network (KAN) as the implicit neural representation (INR) backbone demonstrates a solid theoretical motivation, leveraging KAN’s superior approximation capabilities and parameter efficiency.

2. The method is evaluated on multiple benchmark datasets (KITTI, ScanNet, and 8iVFB).

3. The paper maintains good readability and technical clarity throughout.

**Weaknesses:**

1. The proposed model dictionary appears dataset-specific and may not generalize well to unseen point cloud distributions. Please discuss the potential limitations and any strategies to improve generalization.

2. “we divide the original space S into 2M × 2M × 2M coarse-grained cubes” — It is unclear how boundary issues are handled during reconstruction. Given the independent nature of neighboring blocks, the surface reconstruction may not be seamless. Did the authors observe any boundary artifacts or discontinuities? An additional experimental result illustrating this issue would strengthen the discussion.

3. “For coarse-grained control, we select an optimal model architecture using a pre-computed Pareto frontier that profiles the trade-off between model size and bitrate” — The analysis related to the Pareto frontier is a key design element, but is not clearly presented. Please include the corresponding experiments or plots in the results section to substantiate this claim.

4. A comparison with conventional backbones would better highlight the advantages and significance of LeAFNet.

5. The notation should be made consistent throughout the manuscript (e.g., V - X vs. V\X). Inconsistent notation can confuse readers.

6. Qualitative results should be included to visually demonstrate the reconstruction quality in comparison with existing methods.

**Questions:**

Mentioned in the weaknesses section.

---

> ### Author Response · Authors · 2025-11-12
> **Response to mYum (Part I)**
>
> We thank you for the reviewer mYum’s valuable comments. Our detailed responses to each weakness are provided below.
>
> 1. The proposed model dictionary $\mathcal{M}$ is constructed on 8iVFB-longdress and contains 36 models with different hidden layer dimensions $d$ and depths $l$. The smallest and largest models differ by a factor of 50 in the number of parameters, which allows $\mathcal{M}$ to cover most point cloud compression scenarios.
>    - We tested it on KITTI, ScanNet, and other point clouds from 8iVFB. As shown in Table 1, the $\mathcal{M}$ exhibits good empirical generalization and achieves consistent improvement over other baselines, approaching the current state-of-the-art codec-based DL method Unicorn.
>    - **Potential Limitation:** Considering that we convert point clouds into spatial voxels and apply unified normalization to reduce generalization difficulty, we believe the potential limitation might only arise in extremely sparse or noisy point clouds.
>    - **Potential Solution:** Currently, our model selection strategy within the dictionary only considers the target compression rate (measured in bpp). In the future, we plan to construct model dictionaries that also account for the inherent attributes of the point clouds (e.g., sparsity, sampling method), enabling model configuration and parameterization to depend on both compression targets and object properties. This is indeed a promising direction and could potentially be combined with NAS-like methods.
>
> 2. In Equations 8 and 9, we describe dividing the original space into several coarse-grained cubes. However, please rest assured that this will not lead to boundary inconsistencies.
>    - In our main experiments, we do not perform cube-wise compression for each coarse-grained cube. Instead, this process in Equations 8 and 9 is used to contract the originally sparse sampling space $\mathcal{S}$ into a relatively dense space $\mathcal{V}$, maintaining a balance between positive and negative samples to ensure training stability.
>    - Regarding boundary artifacts or discontinuities, please note that our method constructs a continuous probabilistic field in space through an INR, which prevents noticeable boundary artifacts. We will include qualitative visualization results in the revised PDF.
>    - *In fact, when applying our method to extremely large-scale point cloud compression, we do adopt a cube-wise compression approach which provides better generalization than codec-based methods. We have indeed observed discontinuities between cubes in this case. To address this, we select partially overlapping regions between cubes for separate compression and replace discontinuous boundary areas during decompression according to a designed consistency metric. This part is not included in the current paper as it specifically targets multi-INR collaboration for large-scale point cloud compression.*
>
> 3. We use the Pareto front to construct the model dictionary and select suitable models based on the target compression rate. We conducted a simple experiment to illustrate the dictionary construction process, which is shown in Section 4.3.1 (to be corrected to Section 4.4) and Figure 2. For single point cloud compression, we apply the same compression pipeline using different models. The results show that for different target compression rates (or bpp), choosing different models yields different PSNR values. Therefore, following the Pareto principle, we include in the dictionary the models that achieve the best compression performance under each bpp condition.

---

> ### Author Response · Authors · 2025-11-12
> **Response to mYum (Part II)**
>
> 4. The baseline NeRC used in Table 1 also employs MLP as the backbone. Our method achieves significant improvements over it. The improvement comes not only from backbone differences but also from training details. To demonstrate this, we conducted ablation studies on geometry compression of 8iVFB, as shown below. It can be observed that LeAFNet and the model dictionary bring significant improvements to our method.
>    - NeRC
>    - PICO-MLP (LeAFNet replaced with MLP, w/o the model dictionary)
>    - PICO-MLP-dict (LeAFNet replaced with MLP, w/ the model dictionary)
>
>     | 8iVFB | longdress | loot | redandblack | soldier | Average |
>     |-------|------------|------|--------------|----------|----------|
>     | NeRC | -33.13 / 2.80 | -45.53 / 4.56 | -48.12 / 4.92 | -43.55 / 4.36 | -42.58 / 4.16 |
>     | PICO-MLP | -19.37 / 1.54 | -27.97 / 3.03 | -26.37 / 2.52 | -23.11 / 2.65 | -24.20 / 2.43 |
>     | PICO-MLP-dict | -16.54 / 0.72 | -23.87 / 1.63 | -24.84 / 1.07 | -19.24 / 1.47 | -16.12 / 1.22 |
>
>
> 5. Regarding notation consistency, thank you for pointing this out. We will correct the error in the revised PDF.
>
> 6. Thank you for your comment. We will update this section in the revised PDF after collecting all reviewers’ feedback.
>
> We believe these responses address the reviewers’ concerns. Please feel free to let us know if there are any further questions.

---

### Meta-Review · Area_Chair_D2qD · 2026-01-04

**Summary:**

This paper proposes an INR-based point cloud compression framework built on a KAN-inspired backbone with a model dictionary for explicit rate control. Reviewers found the paper clearly written and technically sound at a high level, with a systematic framework design and evaluation on standard benchmarks. However, the contribution was generally viewed as incremental, with concerns regarding novelty, practical efficiency, and differentiation from recent state-of-the-art methods. While the rebuttal improves clarity and adds supporting experiments, it does not substantially change the overall assessment of the paper’s contribution or applicability. Consequently, I recommend Reject.

**Reviewer Concerns:**

While some presentation issues and empirical gaps were partially addressed by the rebuttal through additional analyses and comparisons, several concrete aspects may still not be fully addressed. First, reviewers questioned the conceptual novelty of the approach, noting that the overall framework and design choices are closely related to existing INR-based point cloud compression methods, with limited contributions beyond incremental integration. Second, the reliance on per-point-cloud training raises practical concerns, as it results in slow encoding and restricts applicability for real-world static point cloud compression. Third, reviewers observed that the advantages over recent state-of-the-art methods such as Unicorn are not clearly demonstrated in terms of overall performance. Finally, the use of a pre-computed Pareto frontier for selecting model architectures raises questions regarding the robustness and generalization of the method to unseen point cloud distributions. Although the rebuttal provides additional explanations and experiments, these issues remain insufficiently addressed.

**Reviewer Scores:**

The paper initially received three borderline reject evaluations. Based on the rebuttal, it is likely that the reviewers’ core concerns are not fully resolved, and the scores are therefore expected to remain unchanged.

---

### Decision · Program_Chairs · 2026-01-26

Reject